# The Effects of Patient-Reported Outcome Screening on the Survival of People with Cancer: A Systematic Review and Meta-Analysis

**DOI:** 10.3390/cancers14215470

**Published:** 2022-11-07

**Authors:** Caterina Caminiti, Giuseppe Maglietta, Francesca Diodati, Matteo Puntoni, Barbara Marcomini, Silvia Lazzarelli, Carmine Pinto, Francesco Perrone

**Affiliations:** 1Clinical and Epidemiological Research Unit, University Hospital of Parma, 43126 Parma, Italy; 2Medical Oncology, Comprehensive Cancer Center, Azienda USL-IRCCS Reggio Emilia, 42122 Reggio Emilia, Italy; 3Clinical Trials Unit, Istituto Nazionale Tumori IRCCS Fondazione G. Pascale, 80131 Naples, Italy

**Keywords:** patient-reported outcome, overall survival, cancer, symptom monitoring, meta-analysis, systematic review

## Abstract

**Simple Summary:**

Patient-reported outcomes (PROs) are information collected directly from patients regarding their health status. Emerging evidence has suggested that integrating PRO assessments into oncology clinical practice can have various benefits for patient care and health. This systematic review and meta-analysis investigated the effects of routine PRO monitoring on the overall survival of people with any type of cancer. We included six studies that compared these interventions to the care that is usually provided to cancer patients. The results seemed to indicate that monitoring PROs in cancer care could positively influence overall survival and that benefits could be largest for individuals with advanced lung cancer. Possible explanations for these findings are that PRO surveillance may allow clinicians to respond to problems more rapidly or that better symptom management could improve tolerance to therapy, thus extending its benefits. However, since available studies are few and of suboptimal quality, additional rigorous research is needed to consolidate our results.

**Abstract:**

This study examined the effects of the routine assessment of patient-reported outcomes (PROs) on the overall survival of adult patients with cancer. We included clinical trials and observational studies with a control group that compared PRO monitoring interventions in cancer clinical practice to usual care. The Cochrane risk-of-bias tools were used. In total, six studies were included in the systematic review: two randomized trials, one population-based retrospectively matched cohort study, two pre–post with historical control studies and one non-randomized controlled trial. Half were multicenter, two were conducted in Europe, three were conducted in the USA and was conducted in Canada. Two studies considered any type of cancer, two were restricted to lung cancer and two were restricted to advanced forms of cancer. PRO screening was electronic in four of the six studies. The meta-analysis included all six studies (intervention = 130.094; control = 129.903). The pooled mortality outcome at 1 year was RR = 0.77 (95%CI 0.76–0.78) as determined by the common effect model and RR = 0.82 (95%CI 0.60–1.12; *p* = 0.16) as determined by the random-effects model. Heterogeneity was statistically significant (I^2^ = 73%; *p* < 0.01). The overall risk of bias was rated as moderate in five studies and serious in one study. This meta-analysis seemed to indicate the survival benefits of PRO screening. As routine PRO monitoring is often challenging, more robust evidence regarding the effects of PROs on mortality would support systematic applications.

## 1. Introduction

Patient-reported outcomes (PROs) are defined by the USA Food and Drug Administration as “any report of the status of a patient’s health condition that comes directly from the patient, without interpretation of the patient’s response by a clinician or anyone else” [1]. PRO measures (PROMs) are derived from the patient self-assessment of a variety of health and wellbeing indices, including measures for health-related quality of life (HRQoL), symptom reporting, satisfaction with care or treatment, economic impacts and the specific dimensions of patient experience, such as depression and anxiety [2,3]. PRO measures are multidimensional and subjective, grounded on patient perceptions and objectively quantified [4]. As PROs can provide crucial information about unique patient experiences during cancer trajectories, their use in clinical practice is becoming increasingly advocated. Subjective patient perceptions can be particularly relevant as it has been shown that patient experiences do not always coincide with clinician understanding [5,6]. These differences in perspective have inspired, for instance, the development of the Patient-Reported Outcomes version of the Common Terminology Criteria for Adverse Events (PRO-CTCAE) by the USA National Cancer Institute [7], which is an internationally accepted system for the grading and reporting of adverse events by clinician [8]. The tool, originally devised for use in clinical cancer trials, is now frequently employed in clinical practice as well and has been translated and cross-culturally adapted into various languages [9].

Various literature reviews have indicated that PRO collection/symptom monitoring in oncology practice can have numerous advantages for patients, including improved communication with healthcare professionals [2,6,10], higher satisfaction [2,11] and higher levels of health-related quality of life [11,12], as well as economic benefits due to decreased emergency room visits and hospital readmissions [11,12]. The growing body of evidence supporting the impacts of PRO detection on patient survival is even more interesting. In a recent systematic review by Lizan et al. [12], five out of six publications assessing this outcome indicated that patient-reported symptom surveillance led to significantly improved survival compared to usual symptom monitoring. Specifically, the review found that active patient-reported monitoring was associated with increased survival for five months or more compared to usual care. However, that review did not provide an assessment of the study quality using appropriate instruments and did not report any meta-analysis data. To our knowledge, no work has yet been published that fills this gap.

Therefore, we conducted a systematic review and meta-analysis to determine whether the assessment of PROs in clinical practice using validated instruments can influence the overall survival of patients affected by any type or stage of cancer.

Review question: Does the use of PROs in oncology clinical practice have an impact on patient survival?

## 2. Materials and Methods

Before conducting this work, the PROSPERO database [13] was searched in April 2022 to identify any existing reviews on the subject in order to avoid replication; however, none were found. This review was designed and conducted following the Preferred Reporting Items for Systematic reviews and Meta-Analyses (PRISMA) guidelines [14]. The protocol was registered with PROSPERO (CRD42022328407) on 10 May 2022.

### 2.1. Search Strategy

Studies were identified by searching the MEDLINE database using the PubMed platform and the Web Of Science Clarivate, with no date or language restrictions. The searches were conducted on 21 June 2022. A “backwards” snowball search was conducted on the references of systematic reviews. The full search strategies and notes on strategy development are provided in the Appendix A.

### 2.2. Study Eligibility

Clinical trials and observational studies with control groups were considered. Studies had to compare the use of a PROM as an intervention in cancer clinical practice to not using a PROM. Any measure that qualified as a PROM according to the aforementioned FDA definition [1] was eligible, provided that it was detected using a validated screening tool that was administered in any format. Comparison had to be to usual care, i.e., the care that is normally provided at the studied center. Thus, we excluded uncontrolled studies, validation studies and studies using PROMs to evaluate another intervention (e.g., PRO data used to measure treatment benefits or risks in medical product clinical trials), as well as studies comparing PROM intervention modalities. Reviews, editorials, commentaries, methodological articles and case reports, along with duplicates/replicates of studies, were excluded.

### 2.3. Population Eligibility

The review concerned adult individuals with any type of cancer in any setting and in any phase of their care trajectory (currently receiving cancer treatment or in follow-up). Studies focused on children (<18 years) were not considered.

### 2.4. Selection Process

Two reviewers independently performed the initial title and abstract screening for relevance to this review using the Rayyan platform [15], which allowed the recording of any discrepancies and the reaching of a consensus. Next, the two reviewers independently examined the full texts of the screened publications and identified eligible papers to include in the review. Any disagreements were resolved by a third independent reviewer.

### 2.5. Data Extraction

Two reviewers independently extracted data from the selected studies using a Microsoft Excel form and disagreements were resolved through discussions, involving a third reviewer when necessary. The extracted data items included title and first author, country, number of centers, cancer type, number of patients, phase of care (active treatment or follow-up), intervention delivery method, screened PROMs and corresponding instruments, estimates of the effects and measures of variability (standard errors or confidence intervals).

Study investigators were contacted when data confirmation was needed.

### 2.6. Risk of Bias Assessment

The internal validity (risk of bias) of the included studies was assessed using the two most recommended tools for interventional studies, according to the design in [16], namely, the Cochrane risk of bias tool for randomized trials (RoB 2) [17] and the Risk Of Bias In Non-randomized Studies of Interventions (ROBINS-I) [18].

RoB 2 [17] is structured into five bias domains, which address all important mechanisms by which bias can be introduced into the results of a trial. They cover the randomization process, deviations from intended interventions, missing outcome data, the measurement of the outcomes and the selection of the reported results. Within each domain, users answer one or more signaling questions. These answers lead to a judgment of “low risk of bias”, “some concerns” or “high risk of bias”. The judgments within each domain lead to an overall judgment for the risk of bias in the results being assessed.

ROBINS-I [18] considers seven domains through which bias can be introduced into non-randomized studies of interventions (NRSIs), covering the confounding and selection of participants into the studies, the classification of the interventions themselves, issues arising after the start of the interventions, biases due to deviations from intended interventions, missing data, the measurement of the outcomes and the selection of the reported results. Responses to “signaling questions” provide the basis for domain-level judgments about the risk of bias, which then provide the basis for an overall risk of bias judgment for particular outcomes using the categories of a “low risk”, “moderate risk”, “serious risk” and “critical risk” of bias.

To present the results of this assessment in a graphical format, we used a traffic light plot to depict the domain level judgments for each study and a summary bar plot figure to show the proportion of studies with a given risk of bias within each domain, weighted by inverse variance. To provide a combined representation of the judgments obtained using the two selected tools, ROBINS-I was used as a reference (seven domains). Thus, for RCTs, the two domains that were not applicable (“selection bias” and “classification of intervention”) were highlighted in gray.

Two reviewers independently applied the tools to each included study and recorded the supporting information and justifications for the judgments of risk of bias for each domain. Doubts were resolved through discussions.

Following the instrument indications, the overall risk of bias was judged according to the following criteria: “low” when all domains were rated as low risk; "some concerns/moderate” when at least one domain was rated as having some concerns/moderate risk but no domain was rated as having a high risk; “high/serious” when at least one domain was rated as having a high/serious risk or if the study was judged to have some concerns for multiple domains in a way that substantially lowered confidence in the results.

### 2.7. Data Synthesis

To summarize the effects of the interventions, the risk ratio (RR) was estimated using the raw data. We performed random-effects meta-analyses using the Paule and Mandel method for the estimation of between-study variance [19,20]. Due to the great variability between the selected studies in terms of sample size, we assigned weights using an inverse variance matrix. The confidence intervals of the overall effects on survival were adjusted by applying the Hartung–Knapp–Sidik–Jonkman (HKSJ) approach [21,22,23] to account for the uncertainty in the variance estimates. I^2^ statistics tests were calculated to quantify the degree of study heterogeneity [24]. The I^2^ value that established significant heterogeneity was 70%. The level of significance was set at *p* < 0.050.

We did not perform formal subgroup analyses due to the insufficient number of included studies. Furthermore, we planned to assess publication bias using funnel plot representation and a Peter’s test at a 10% level; however, this was not possible because fewer than 10 studies were considered.

The data were processed using R statistical software (R: a language and environment for statistical computing; the R Foundation for Statistical Computing, Vienna, Austria), v. 4.0.3, with the meta and metasens packages [25].

### 2.8. Patient and Public Involvement

Patients nor the public were involved in this research.

## 3. Results

### 3.1. Study Selection

A total of 3723 articles were retrieved from the two databases and uploaded into the Rayyan platform. After removing duplicates, 2433 records underwent the title and abstract screening. We chose not to apply automation tools to determine ineligibility in order to increase accuracy; thus, all references were screened manually. In total, 14 reports were identified as potentially eligible and underwent a full text review. Of these, six [26,27,28,29,30,31] were excluded, mainly because the outcomes of interest in this review were not measured or because the types of intervention or study aims were not eligible for our study question (Appendix A). Overall, six studies [32,33,34,35,36,37] were included in our systematic review, to which two follow-up publications [38,39] were added for the meta-analysis.

A flow diagram depicting the selection process is provided in Figure 1.

### 3.2. Study Characteristics

The characteristics of the six studies included in the review are shown in Table 1.

Three out of the six studies [32,35,37] were multicenter. Two of the six studies [34,35] were set in Europe, three [33,36,37] were set in the USA and one [32] was set in in Canada. Two studies were RCTs [33,35], one was a population-based retrospectively matched cohort analysis [32], two used a pre–post design with historical controls [36,37] and one was a non-randomized controlled trial in which the controls were patients who refused the intervention [34].

Regarding included the populations, three out of the six studies [33,34,35] restricted eligibility to patients receiving active treatment, while the others recruited patients in any phase of care. Two studies [34,35] focused exclusively on lung cancer, two [33,36] focused on advanced cancers and two [32,37] included any type and stage of cancer.

For PRO screening instruments, three studies [32,36,37] used ESAS (Edmonton Symptom Assessment System), one study [37] screened for depression using the 9-item Patient Health Questionnaire (PHQ-9), one trial [33] employed STAR (Symptom Tracking and Reporting), one study [34] used items from the NCI’s PRO-CTCAE (Patient-Reported Outcomes version of the Common Terminology Criteria for Adverse Events) and one study [35] used the e-FAB (e-follow-up) application. The mode of administration was electronic in all studies except for those by Patel et al. [36,37], in which the interventions were telephonic.

The meta-analysis considered two additional papers that reported follow-up mortality data for the two RCTs [38,39]: Basch 2017 [38] reported the results of a preplanned post-hoc analysis and Denis 2019 [39] described the results of a 2-year follow-up after the study was stopped early. Overall, 130.094 patients in the intervention arms vs. 129.903 in the control arms were considered for the meta-analysis, of whom 124,259 (48%) were women.

### 3.3. Impact of Patient-Reported Outcome Monitoring on Overall Survival

All six studies selected for the systematic review were included in the meta-analysis as they all reported the necessary raw mortality data. The two non-randomized studies with historical controls [36,37] (611 patients in the intervention arms vs. 509 patients in the control arms) reported survival after 1 year as a secondary output, the population-based retrospectively matched cohort study [32] (128,893 patients in the intervention arm vs. 128,893 patients in the control arm) analyzed survival up to 5 years as a primary outcome and the non-randomized controlled study [34] (89 patients in the intervention arm vs. 115 patients in the control arm) analyzed survival up to 4 years as a secondary outcome. Two RCTs [33,35] (overall, 501 patients in the intervention arms vs. 386 patients in the control arms) reported survival results from post-hoc analyses [38,39] as a secondary outcome with a median follow-up of 7 years and the other reported the results as a primary outcome after 2 years.

Figure 2 displays the results of our meta-analysis, together with the risk of bias assessments described in the following section.

All studies except for the first study by Patel et al. [36] demonstrated reductions in mortality, which were statistically significant in four studies [32,33,34,35] with RRs ranging between 0.49 to 0.79. The pooled mortality outcome after 1 year (the observation timeframe that was common to all studies) was RR = 0.77 (95%CI 0.76–0.78) as determined using the common effect model and RR = 0.82 (95%CI 0.60–1.12; *p* = 0.16) as determined using the random-effects model. Heterogeneity was statistically significant (I^2^ = 73%; *p* < 0.01).

### 3.4. Quality of Included Studies

In addition to the forest plot, Figure 2 depicts both a traffic light plot showing the risk of bias judgments for the individual domains for each study and the overall risk of bias and a summary bar plot showing the cumulative risk of bias percentage for each domain. The overall risk of bias was judged to be moderate for all studies, except for one of the Patel studies [36], which was rated as having a serious risk. This was mainly due to selection bias as the intervention group comprised a higher number of patients with baseline stage IV disease than the control group. The most frequent problem, which was present in five out of the six studies (corresponding to a weighted percentage of 78.8%), concerned bias in the selection of the reported results; specifically, findings on survival were not reported in detail or only referred to patient subgroups. The reasons for the judgments for each bias domain are provided in the Appendix A.

### 3.5. Further Analyses

The additional outcomes that were planned in the protocol (disease-free survival, progression-free survival and event-free survival) could not be measured because they were not investigated in the included studies.

Since two of the six included studies [34,35] focused on patients with advanced lung cancer, an exploratory subgroup analysis was performed for this cancer type. The results are depicted in the Appendix A.

Finally, we carried out a sensitivity analysis, excluding the study with the serious overall risk of bias [36]. The overall survival determined using the random-effects model was RR = 0.78 (95%CI = 0.63–0.95; *p* = 0.02; I^2^ = 51%; *p* = 0.08) (Appendix A).

## 4. Discussion

To our knowledge, this is the first meta-analysis of the effects of PRO monitoring in oncology practice in terms of overall patient survival. The results seemed to indicate that monitoring patient-reported outcomes in clinical practice could have a positive impact on the overall survival of people with cancer. Specifically, our overall estimate indicated an 18% reduction in the risk of death, although this effect was not statistically significant and adjustment for confounding factors was not possible. Furthermore, we wished to perform a formal subgroup analysis for lung cancer patients that included multiple studies; however, the two authors we contacted could not provide us with the necessary data. The exploratory sub-analysis we were able to perform, which was limited to two studies, suggested that patients with advanced lung cancer might benefit the most from these interventions.

The systematic introduction of PRO monitoring into clinical practice is often difficult due to operational and financial barriers to the implementation of these complex interventions, as well as the uncertainty among physicians regarding their usefulness in actual practice [3,40]. Therefore, it is important to identify the categories of patients who are most likely to benefit and consequently focus efforts on these populations [41].

Survival improvements after PRO surveillance are plausible, although the mechanisms by which this important benefit is achieved remain debatable [12,42]. One hypothesis is that patient-reported surveillance allows doctors to respond to problems earlier, thus preventing complications, unexpected hospitalizations or the discontinuation of chemotherapy. Another possibility is that improvements in symptom management may also allow patients to tolerate their symptoms better and, consequently, benefit from chemotherapy for a longer time than when receiving usual care. Finally, the systematic collection of PROs can support the recognition of problematic symptoms, thus promoting patient empowerment and self-management.

The results of this work must be interpreted with caution, due to both the lack of statistical significance of the pooled estimate and the suboptimal quality of the six included studies (none of which were rated as having a low risk of bias). We also observed an under-representation of women in three of the studies (Demedts et al. [34] (24%), Denis et al. [35] (33%) and Patel et al. [37] (41.5%)), as well as the absence of any formal analyses of gender differences in all studies. Given these limitations, more rigorous research is needed to consolidate the positive signs yielded by this work.

## 5. Conclusions

In conclusion, the available evidence was insufficient to draw firm conclusions that PRO monitoring could extend survival. Further evidence is expected to emerge from PRO-TECT, a large randomized cluster study that is currently underway in 52 community oncology practices in the United States of America on 1191 subjects with metastatic cancer [43]. The results regarding overall survival as the primary outcome are not yet available. Therefore, we intend to update this meta-analysis after the publication of those results.

In any case, as indicated in the European guidelines, there is substantial evidence to support the benefits and feasibility of implementing PROMs in clinical outpatient cancer care, particularly for patients receiving active therapy or during the observation of therapy with a high risk of recurrence [41]. Furthermore, routine PRO surveillance could help to standardize clinical care in a world in which the volume of patients is increasing and ensure patient engagement during the entirety of their cancer trajectory.

## Figures and Tables

**Figure 1 cancers-14-05470-f001:**
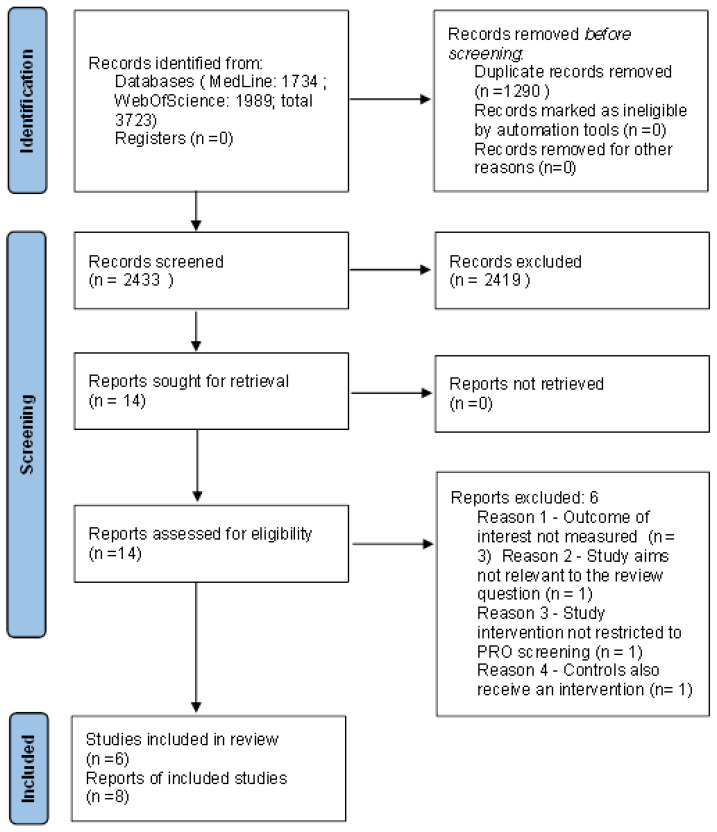
A PRISMA 2020 flow diagram of the process of identifying studies (both included and excluded).

**Figure 2 cancers-14-05470-f002:**
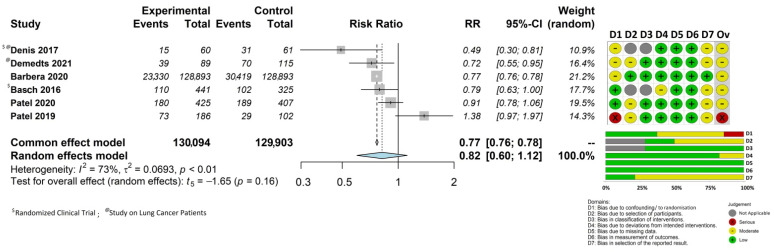
A forest plot of the survival pooled effect and a summary of the risk of bias (traffic light plot and summary bar plot) [32,33,34,35,36,37].

**Table 1 cancers-14-05470-t001:** Characteristics of included studies.

First Author	Year	Study Design	Country	Single Center or Multicenter	No. of Patients	Age (Years)	Sex (% Female)	Cancer Type	Phase of Care (Active Treatment vs. Follow-Up)	Intervention vs. Control	PRO Instrument Used for Intervention	Mode of Administration	Follow-up Length for Survival	Survival as Primary or Secondary Outcome?
Barbera	2020	Population-based retrospectively matched cohort analysis	Canada	Multicenter	257,786 (128,893 patients with ESAS exposure matched to 128,893 patients without ESAS exposure)	Mean: 64 SD: 13	47.8%	Various	Any	12-month telephonic screening: high-risk patients were called weekly and low-risk patients were called monthly. Historical controls received usual cancer care without standardized symptom screening or management.	Edmonton Symptom Assessment System (ESAS)	Electronic (Touch screen at the center)	5 years (median: 1.4)	Primary
Basch	2016	Randomized controlled trial	USA	Single center	766 (441 intervention vs. 325 control)	Median: 61 Range: 26–91	58%	Advanced solid tumors: metastatic breast, genitourinary, gynecologic or lung cancers	Active treatment	Patients were randomly assigned to either report 12 common symptoms via tablet computers or receive usual care consisting of symptom monitoring at the discretion of clinicians. Those with home computers received weekly email prompts to report symptoms between visits. Treating physicians received symptom printouts at visits and nurses received email alerts when participants reported severe or worsening symptoms.	Symptom Tracking and Reporting (STAR)	Electronic (web-based)	Median follow-up of 7 years (interquartile range 6.5–7.8) ^$^	Secondary
Demedts	2021	Non-randomized controlled study	Belgium	Single center	204 stage IV non-small cell lung cancer patients (89 intervention vs. 115 control)	Median: 66 Range: 32–88	24%	Lung cancer	Active treatment	Patients were invited by email every week to report on the side effects of their treatment. Feedback loops were created with automatically triggered electronic alerts to the care team when a predefined threshold of symptoms was reached. When patients refused, usual care was offered.	Items from the Patient-Reported Outcomes version of the Common Terminology Criteria for Adverse Events (PRO-CTCAE) library	Electronic (email reminders and digital reporting)	4 years	Secondary
Denis	2017	Randomized controlled trial	France	Multicenter	133 patients enrolled: 12 deemed ineligible after randomization and 121 retained in the intent-to-treat analysis (60 intervention vs. 61 control)	Median: 64.5 Range: 35.7–88.1	33%	Advanced-stage lung cancer	Active treatment	Personalized follow-up strategy based on 12 symptoms that were self-scored weekly and transmitted to the oncologist. Clinical follow-ups in both arms included oncology visits at least every 3 months.	e-Follow-up Application (e-FAP)	Electronic (web-based)	2 years *	Primary
Patel	2019	Non-randomized study with historical control	USA	Single center	288 (186 intervention vs. 102 control)	Mean: 79 SD: 8	55%	Advanced cancers	Any	12-month telephonic program in which a lay health worker (LHW), supervised by a physician assistant (PA), assessed patient symptoms after diagnosis, with the frequency of symptom screening varying on the basis of patient risk (once a week for high-risk patients and at least once a month for low-risk patients). All participants received usual cancer care.	Edmonton Symptom Assessment System (ESAS)	Telephone	1 year	Secondary
Patel	2020	Non-randomized study with historical control	USA	Multicenter	832 (425 intervention vs. 407 control)	Mean: 79 SD: 8.3	41.5%	Various new diagnoses of solid or hematologic malignant neoplasms	Any	12-month telephonic program in which a lay health worker (LHW), supervised by a physician assistant (PA), assessed patient symptoms after diagnosis, with the frequency of symptom screening varying on the basis of patient risk (once a week for high-risk patients and at least once a month for low-risk patients). All participants received usual cancer care.	For symptoms: Edmonton Symptom Assessment System (ESAS). For depression: the 9-item Patient Health Questionnaire (PHQ-9)	Telephone	1 year	Secondary

^$^ Results for the 7-year follow-up are reported in the following article on the same study [37] (Basch 2017), * Results for the 2-year follow-up are reported in the following article on the same study [39] (Denis 2019).

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
