# Peer review of "The Effects of Patient-Reported Outcome Screening on the Survival of People with Cancer: A Systematic Review and Meta-Analysis"

_cancers, 2022, doi:10.3390/cancers14215470_

Round 1

Reviewer 1 Report

The authors have chosen an extremely "hot topic", which will be much debated in the coming future. Integrating PRO's into the shared decision making will be critical, specifically also in light of the definition of quality with respect to VBHC.

Whether PROs add a survival benefit is a much debated open question, separating into believers and non-believers. These authors chose to add the strongest evidence possible performing a systematic review. Their data indicated that reduction of risk of death may be up to 18%. There is hardly no other treatment modality known which may add such additional benefit at all, underlining the importance to further explore on PROs. 

The greatest limitation as expected is the quality of data available in the literature. Reason why health care needs to focus more on data quality, and  the introduction of PRO will helping a lot in this respect.

In my eyes, this is a landmark paper which needs to be published. It is very nicely written and very carefully worded, specifically reporting on the findings in the abstract as well as in the conclusion.

Author Response

We thank the reviewer for taking the time to carefully read our paper and for his/her appreciation of our work and of its purpose. We completely agree that data quality is a serious limitation, especially in the field of complex interventions like these, which also involve assessment of subjective patient experience. That is why we intend to update this meta-analysis when new evidence emerge, in the hope that future trials may be more rigorously conducted and thus provide more robust data.

Reviewer 2 Report

As a clinician, i consider this study very relevant. It shows how a systematic interrogatory during treatment or different phases of the journey of a patient with cancer can make a difference in medical care. 

There are some human variables as fatigue, routine that can make the medical doctor not ask for every symptom that at the end is very significative. 

Analyzing the methodological presentation i appreciate the Cochrane image of bias which shows that even if the number of studies included is short the quality of information is high. 

I consider this tools must be generalized in medical practice in order to standardized clinical care in a world where high volume of patients is increasing. 

The conclusions seems simple, i suggest to improve the idea of which are the potential use of these tools in a more general way. 

Author Response

We thank the reviewer for taking the time to evaluate our manuscript, and for the interesting and useful comments on the values that PRO screening may have for a clinician. Since implementation of these demanding interventions ultimately falls upon clinicians, it is essential to understand and emphasize the benefits from their point of view. Therefore, we are grateful for suggesting to enrich our conclusions accordingly.